# NDAT Targets PI3K-Mediated PD-L1 Upregulation to Reduce Proliferation in Gefitinib-Resistant Colorectal Cancer

**DOI:** 10.3390/cells9081830

**Published:** 2020-08-03

**Authors:** Tung-Yung Huang, Tung-Cheng Chang, Yu-Tang Chin, Yi-Shin Pan, Wong-Jin Chang, Feng-Cheng Liu, Ema Dwi Hastuti, Shih-Jiuan Chiu, Shwu-Huey Wang, Chun A. Changou, Zi-Lin Li, Yi-Ru Chen, Hung-Ru Chu, Ya-Jung Shih, R. Holland Cheng, Alexander Wu, Hung-Yun Lin, Kuan Wang, Jacqueline Whang-Peng, Shaker A Mousa, Paul J. Davis

**Affiliations:** 1Graduate Institute of Cancer Biology and Drug Discovery, College of Medical Science and Technology, Taipei Medical University, Taipei 11031, Taiwan; charvel0203@gmail.com (T.-Y.H.); extraganoderma@gmail.com (Y.-S.P.); wjchang@tmu.edu.tw (W.-J.C.); lizilin0919@gmail.com (Z.-L.L.); aquarlus9132@gmail.com (Y.-R.C.); a0918362166@tmu.edu.tw (H.-R.C.); shihyj@tmu.edu.tw (Y.-J.S.); jqwpeng@nhri.org.tw (J.W.-P.); 2Graduate Institute of Nanomedicine and Medical Engineering, College of Medical Engineering, Taipei Medical University, Taipei 11031, Taiwan; wangk007@gmail.com; 3Division of Colorectal Surgery, Department of Surgery, Taipei Medical University Shuang Ho Hospital, New Taipei City 235041, Taiwan; roussekimo@yahoo.com.tw; 4Division of Colorectal Surgery, Department of Surgery, School of Medicine, College of Medicine, Taipei Medical University, Taipei 11031, Taiwan; 5School of Dentistry, Taipei Medical University, Taipei 11031, Taiwan; yutangchin@gmail.com; 6Division of Rheumatology, Immunology, and Allergy, Tri-Service General Hospital, Taipei 114, Taiwan; lfc10399@gmail.com; 7School of Pharmacy, College of Pharmacy, Taipei Medical University, Taipei 11031, Taiwan; hastuti.ema.d@gmail.com (E.D.H.); sjchiu@tmu.edu.tw (S.-J.C.); 8Department of Biochemistry and Molecular Cell Biology, College of Medicine, Taipei Medical University, Taipei 11031, Taiwan; shwu@tmu.edu.tw; 9Core Facility Center, Department of Research Development, Taipei Medical University, Taipei 11031, Taiwan; austinc99@tmu.edu.tw; 10TMU Research Center of Cancer Translational Medicine, Taipei Medical University, Taipei 11031, Taiwan; 11Department of Molecular and Cellular Biology, College of Biological Sciences, University of California, Davis, CA 95616, USA; rhch@ucdavis.edu; 12The Ph.D. Program for Translational Medicine, College of Medical Science and Technology, Taipei Medical University, Taipei 110, Taiwan; 13Graduate Institute for Cancer Molecular Biology and Drug Discovery, College of Medical Science and Technology, Taipei Medical University, Taipei 11031, Taiwan; 14Integrated Laboratory, Center of Translational Medicine, Taipei Medical University, Taipei 11031, Taiwan; 15Cancer Center, Wan Fang Hospital, Taipei Medical University, Taipei 11031, Taiwan; 16Traditional Herbal Medicine Research Center of Taipei Medical University Hospital, Taipei Medical University, Taipei 11031, Taiwan; 17Pharmaceutical Research Institute, Albany College of Pharmacy and Health Sciences, Albany, NY 12208, USA; Shaker.Mousa@acphs.edu (S.A.M.); pdavis.ordwayst@gmail.com (P.J.D.); 18Department of Medicine, Albany Medical College, Albany, NY 12208, USA

**Keywords:** colorectal cancer (CRC), gefitinib, NDAT, PD-L1, PI3K

## Abstract

The property of drug-resistance may attenuate clinical therapy in cancer cells, such as chemoresistance to gefitinib in colon cancer cells. In previous studies, overexpression of PD-L1 causes proliferation and metastasis in cancer cells; therefore, the PD-L1 pathway allows tumor cells to exert an adaptive resistance mechanism in vivo. Nano-diamino-tetrac (NDAT) has been shown to enhance the anti-proliferative effect induced by first-line chemotherapy in various types of cancer, including colorectal cancer (CRC). In this work, we attempted to explore whether NDAT could enhance the anti-proliferative effect of gefitinib in CRC and clarified the mechanism of their interaction. The MTT assay was utilized to detect a reduction in cell proliferation in four primary culture tumor cells treated with gefitinib or NDAT. The gene expression of *PD-L1* and other tumor growth-related molecules were quantified by quantitative polymerase chain reaction (qPCR). Furthermore, the identification of PI3K and PD-L1 in treated CRC cells were detected by western blotting analysis. PD-L1 presentation in HCT116 xenograft tumors was characterized by specialized immunohistochemistry (IHC) and the hematoxylin and eosin stain (H&E stain). The correlations between the change in PD-L1 expression and tumorigenic characteristics were also analyzed. **(3)** The *PD-L1* was highly expressed in Colo_160224 rather than in the other three primary CRC cells and HCT-116 cells. Moreover, the *PD-L1* expression was decreased by gefitinib (1 µM and 10 µM) in two cells (Colo_150624 and 160426), but 10 µM gefitinib stimulated *PD-L1* expression in gefitinib-resistant primary CRC Colo_160224 cells. Inactivated PI3K reduced *PD-L1* expression and proliferation in CRC Colo_160224 cells. Gefitinib didn’t inhibit *PD-L1* expression and PI3K activation in gefitinib-resistant Colo_160224 cells. However, NDAT inhibited PI3K activation as well as PD-L1 accumulation in gefitinib-resistant Colo_160224 cells. The combined treatment of NDAT and gefitinib inhibited pPI3K and PD-L1 expression and cell proliferation. Additionally, NDAT reduced PD-L1 accumulation and tumor growth in the HCT116 (*K-RAS* mutant) xenograft experiment. **(4)** Gefitinib might suppress *PD-L1* expression but did not inhibit proliferation through PI3K in gefitinib-resistant primary CRC cells. However, NDAT not only down-regulated PD-L1 expression via blocking PI3K activation but also inhibited cell proliferation in gefitinib-resistant CRCs.

## 1. Introduction

In recent years, colorectal cancer (CRC) has been recognized as the third most prevalent and fourth most lethal cancer worldwide [1]. The incidence of this cancer has rapidly increased in the past 10 years in Taiwan (4576 cases/10^6^ population in 2002 vs. 9299 cases/10^6^ population in 2012) [2]. New therapeutic approaches for metastatic colon cancer are needed. Mutations of *Adenomatous Polyposis Coli* (*APC*), *K-RAS*, and *β-catenin* genes have been proposed as early events in the tumorigenesis of CRC [3,4], but whether relationships exist among such events is unclear. The *APC*-mediated initiation of intestinal tumorigenesis requires normal epidermal growth factor receptor (EGFR) activity for the establishment of intestinal tumors [5]. Aberrant activation of EGFR stimulates several intracellular signaling pathways—Phosphoinositide 3-kinase (PI3K)/AKT, RAS/RAF/MEK/ERK, Src/signal transducer and activator of transcription (STAT)—Which in turn cause augmented cell proliferation and other oncogenic characteristics in cancers [6]. On the other hand, low EGFR expression in CRC cells is correlated clinically with low tumor metastasis risk and better survival [4,7].

In addition to EGFR, thyroid hormone (thyroxine, T_4_) has been shown to be involved in CRC progression [8,9,10,11]. Thyroid hormone can also induce expression/activation of the programmed cell death protein 1 (PD-1)/programmed death-ligand 1 (PD-L1) immune checkpoints in different types of cancers [10,12]. PD-L1 expression has a positive correlation with cancer progression and decreased patient survival [13,14,15]. PD-L1 can be induced by EGF [16] and by interferon-γ (IFN-γ) [17]. PD-1/PD-L1 is an essential regulator of the interactions between T cells and tumor cells [18,19,20] and protects tumor cells against immune system-mediated destruction (apoptosis). Mechanisms involved in thyroid hormone-induced expression of PD-L1 are not fully understood. However, the thyroid hormone enhances oxidation, a consequence of which is constitutive of PD-L1 expression. In addition, the activation of ERK1/2 is required for T_4_-induced PD-L1 expression [12], and the activation of the PI3K/AKT pathway regulates the expression of PD-L1 in triple-negative breast cancer cells [21].

Gefitinib is a tyrosine kinase inhibitor (TKI) anticancer agent with multiple mechanisms of action. It promotes cell cycle arrest and decreases the expression of cancer metastasis-related proteins, such as basic fibroblast growth factor (bFGF) and matrix metalloproteinases-2 (MMP-2) and MMP-9 [22]. Gefitinib has been combined with other chemotherapeutic agents to manage a variety of cancers [23,24,25,26]. On the other hand, a deaminated L-thyroxine analog, tetraiodothyroacetic acid (tetrac), and its nanoparticulate analog nano-diamino-tetrac (NDAT) have been shown to inhibit proliferation of cancer cells in vitro and in vivo [8,9,10,27,28,29]. The anticancer properties of NDAT include inhibition of cancer cell proliferation, angiogenesis, metastasis, and immune-escape mechanisms of cancer cells, and it induces apoptosis of cancer cells [27,28,30,31]. In addition, NDAT has been shown to inhibit immune-escape mechanisms of cancer cells [12]. Recently, we demonstrated that NDAT enhances gefitinib-induced anti-proliferation in *K-RAS* mutant CRC HCT116 cells in vitro and in murine xenografts [9]. However, the mechanisms involved in the potentiating effect of NDAT on gefitinib-induced anticancer activity in xenografts and primary CRC cell lines have not been defined.

In the current study, we investigated the mechanisms of NDAT-induced anti-proliferation in gefitinib-resistant primary CRC cell cultures and xenografts. We studied the role of activated PI3K on the PD-L1 expression and cancer growth in CRC primary cultures and *K-RAS* mutant HCT116 cell xenografts. The results indicated that the inactivation of PI3K by NDAT or the PI3K inhibitor was able to inhibit PD-L1 accumulation and cell proliferation. On the other hand, gefitinib didn’t inhibit *PD-L1* expression in gefitinib-resistant primary CRC cells. Therefore, NDAT may be useful to compensate for the therapeutic effect in gefitinib-resistant patients.

## 2. Materials and Methods

### 2.1. Cell Line

Human colorectal cancer cell line HT-29 (ATCC^®^ HTB-38^TM^) and HCT116 (ATCC^®^ CCL-247^TM^) were purchased from the American Type Culture Collection (ATCC, Manassas, VA, USA) by the Bioresource Collection and Research Center (BCRC, Hsinchu, Taiwan) and maintained in RPMI-1640 (Life Technologies Corp., Carlsbad, CA, USA), supplemented with 10% FBS. The incubation conditions were 5% CO_2_ at 37 °C.

### 2.2. Tissue Specimen Source of Primary Cultures of Tumor Cells

CRC patients were admitted to the Division of Colorectal Surgery, Department of Surgery, Shuang-Ho Hospital (Taipei Medical University, Taipei, Taiwan) and were included in this study according to standardized diagnostic criteria. All patients provided informed consent to the protocol approved by the Taipei Medical University Joint Institutional Review Board (TMU-JIRB number: N201603078, duration of validity was from 30 November 2017 to 29 November 2018). Samples of resected CRCs were collected from patients. The enrolled patients received no chemotherapy or radiation therapy prior to surgery. The histopathology of each specimen was carefully evaluated.

### 2.3. Specimen Preparation and Tumor Cell Isolation

The isolation and culture procedures for primary cultures of human CRC cells were modified from previous studies [32,33]. Four primary human CRC cell samples (Colo_150624, Colo_150812-2, Colo_160224, and Colo_160426) were isolated and cultured in RPMI 1640 medium with 10% FBS and antibiotics (penicillin 100 IU/mL, streptomycin 100 μg/mL, amphotericin B 2.5 μg/mL) until use. Before these treatments, cells were placed in serum-free medium for 24 h starvation. The detailed information is described in the Appendix A.

### 2.4. Cell Viability Assay

The four established primary cultures of human CRC cells (Colo_150624, Colo_150812-2, Colo_160224, and Colo_160426) (5 × 10^3^ cells per well) were cultured in 96-well plates, then treated with NDAT (0.01 and 0.1 μM) (NanoPharmaceuticals LLC, Rensselaer, NY, USA), gefitinib (0.1, 1, and 10 μM) (ZD1839; Selleck Chemicals, Houston, TX, USA) and combination treatment for six days. Cell proliferation was examined by the MTT assay as described previously [8,34].

### 2.5. Quantitative RT-PCR (qPCR)

As described previously [8,9], the total RNA was extracted by using the Illustra RNAspin Mini RNA Isolation Kit (GE Healthcare Life Sciences, Buckinghamshire, UK). One μg of DNase I-treated total RNA was reverse-transcribed into cDNA and used as the template for real-time PCR reactions and analysis. The real-time PCR reactions were performed using the QuantiNovaTM SYBR^®^ Green PCR Kit (QIAGEN, Germantown, MD) on a CFX Connect™ Real-Time PCR Detection System (Bio-Rad Laboratories, Inc., Hercules, CA, USA). The primer sequences were listed in Table 1. The fidelity of thre qPCR reaction was determined by melt curve analysis. The relative gene expression (normalized to 18S gene) was calculated by the ΔΔCT method. 

### 2.6. Western Blotting

To test the signaling pathways involved in the anti-proliferative effects of NDAT, gefitinib, and their combination, we applied western blot to quantify the protein expression levels in the total cell lysates of the primary cultures of human CRC cells (Colo_160224 cells); the cells were treated with NDAT, gefitinib, or in combination for 24 h. Protein samples were resolved by 10% sodium dodecyl sulfate-polyacrylamide gel. The resolved proteins were transferred to Millipore Immobilon-PSQ Transfer PVDF membranes (Millipore, Billerica, MA, USA) by the Mini Trans-Blot Cell (Bio-Rad Laboratories, Inc.). The membranes were blocked with 5% skim milk in TBST, and incubated with primary antibodies against PD-L1 and GAPDH (GeneTex International Corp., Hsinchu City, Taiwan) at 4 °C overnight. HRP-conjugated secondary antibodies and the Immobilon TM Western Chemiluminescent HRP Substrate (WBKLS0500, Millipore, Billerica, MA, USA) were used to detect the target antigen. Western blots were visualized and recorded with the BioSpectrum Imaging System (UVP, LLC, Upland, CA, USA). The densitometric analyses of western blots were analyzed with the ImageJ 1.5 software (NIH, Bethesda, MD, USA).

### 2.7. Confocal Microscopy

Exponentially growing primary colorectal cancer cells and HT-29 were seeded on sterilized cover glasses (Paul Marienfeld, LaudaKönigshofen, Germany). After treatment of NDAT (0.1 μM) or LY294002 (10 μM) for 24 h, cells were fixed with 4% paraformaldehyde in phosphate-buffered saline (PBS) for 30 min and then permeabilized in 0.06% Triton X-100 for 30 min. Cells were incubated with monoclonal rabbit anti-PD-L1 antibody, followed by an Alexa-647-labeled goat anti-rabbit antibody (Abcam, Cambridge, MA, USA) and mounted in EverBrite Hardset mounting medium with DAPI (Biotium, Fremont, CA). The fluorescent signals were recorded and analyzed with the TCS SP5 Confocal Spectral Microscope Imaging System (Leica Microsystems). The figures shown are representative of at least four fields for each experimental condition.

### 2.8. Xenografts

Forty nude mice (BALB/cAnN.Cg-Foxn1nu/CrlNarl, male) were purchased from the National Laboratory Animal Center (Taipei, Taiwan) and were housed in a reserved, pathogen-free facility, and were treated following the protocols approved by the Institutional Animal Care and Use Committee of the National Defense Medical Center, Taipei, Taiwan (IACUC-15-340). The detailed information of studies was described previously [8,9] and in the Appendix A.

### 2.9. Immunohistochemical (IHC) Staining

Serial sections of 5 μm thickness were sliced from paraffin-embedded xenograft tissue samples for IHC staining. The IHC procedure was carried out according to the Novolink^TM^ max polymer detection system (RE7280-K, Leica Biosystems Newcastle Ltd). Briefly, antigen-retrieved sections were neutralized with a peroxidase block and blocked with a protein block for 5 min each. The primary antibody, rabbit anti-PD-L1 antibodies (Cell Signaling Technology, Inc., Beverly, MA, USA), 200-folds diluted in PBST containing 1% BSA was added to the sections and incubated at 4 °C overnight. After TBS washing, sections were primed with post-primary and subsequent Novolink polymer for 30 min each. The DAB substrate kit was used to visualize the PD-L1 levels (Ab64238, Abcam, Cambridge, UK). Cell nuclei were counterstained by hematoxylin. Sections were examined under the Nikon Eclipse ci optical microscope imaging system (Nikon Eclipse ci, Nikon Instruments, Tokyo, Japan). Low-power (40×) and high-power (400×) microscopic fields were randomly chosen from each slide to demonstrate the positive staining of PD-L1 with brown staining.

### 2.10. Statistical Analysis

The fold changes in gene expression in qPCR, in western blot protein densities, and in tumor volume were evaluated with the IBM SPSS Statistics software version 19.0 (SPSS Inc., Chicago, IL, USA). Student’s *t-test* was conducted, and changes were considered significant at *p* < 0.05 (*, #, &, $), 0.01 (**, ##, &&, $$) and 0.001 (***, ###, &&&, $$$). One-way analysis of variance (ANOVA) with the Duncan’s *post-hoc* test was used to analyze the difference of basal *PD-L1* expression levels among HCT116 cells and the four established primary cultures of human CRC cells (Colo_150624, Colo_150812-2, Colo_160224, and Colo_160426). 

## 3. Results

### 3.1. NDAT Enhances Gefitinib-Induced Anti-Proliferation in Primary Cultures of Human CRC Cells

NDAT inhibits PD-L1 expression and cell proliferation stimulated by thyroid hormone in cancer cells ** [12,35]**. Gefitinib has been shown to suppress cell proliferation in *K-RAS* wild type HT-29 cells, but not in *K-RAS*-mutant HCT116 cells [9]. NDAT potentiates a gefitinib-induced anti-proliferative effect in both types of cancer cells [9]. We investigated the effects of NDAT and gefitinib in primary CRC cell lines. Gefitinib significantly inhibited cell proliferation in Colo_150624 (Figure 1A) and Colo_160426 cells (Figure 1B) in a concentration-dependent manner although 10 µM gefitinib did not inhibit Colo_160426 cancer growth. On the other hand, gefitinib lacked such activity in Colo_150812-2 (Figure 1C) and Colo_160224 cells (Figure 1D). In the latter, a high concentration of gefitinib (10 μM) significantly enhanced cell proliferation in Colo_160224 cells (Figure 1D), but not in Colo_160426 (Figure 1B) and Colo_150812-2 (Figure 1C) cells. A low concentration of NDAT (0.01 μM) inhibited the proliferation of these primary cultures of CRC cells but not in Colo_150812-2 (Appendix A). The treatment of NDAT (0.1 μM) significantly suppressed cell proliferation and showed a potentiated effect when combined with gefitinib in these cells (Figure 1).

### 3.2. NDAT Enhances Gefitinib-Inhibited PD-L1 Expression in Primary Cultures of CRC Cells

*PD-L1* expression may affect cancer cell proliferation [35,36]; therefore, we examined the possible linkage among PD-L1, cell proliferation, and gefitinib resistance. Gefitinib at a concentration of 10 μM significantly inhibited *PD-L1* expression only in Colo_150624 cells (Figure 2A), whereas it significantly enhanced *PD-L1* expression in Colo_160224 cells (Figure 2D). On the other hand, NDAT (0.1 μM) significantly potentiated gefitinib-repressed *PD-L1* expression in primary cultures of CRC cells (Figure 2). The basal expression levels of *PD-L1* mRNA in HCT116 cells and four established primary cultures of human CRC cells were evaluated. Expression of *PD-L1* in gefitinib-resistant Colo_160224 cells was meaningfully higher than that in other primary CRC cancer cell lines and HCT116 cells (Appendix A). NDAT significantly repressed *PD-L1* expression in these primary cultures of CRC cells at the concentration of 0.1 μM (Figure 2). Expression of PD-L1 was significantly inhibited by 1 μM gefitinib in Colo_150624 (Appendix A), Colo_150812-2 (Appendix A), and Colo_160224 (Appendix A) cells. 

### 3.3. NDAT But Not Gefitinib Functions as PI3K Inhibitor to Inhibit PD-L1 Expression and Cell Proliferation in Gefitinib-Resistant Primary CRC

Constitutive activation of PI3K/Akt and RAS/ERK pathways is associated with gefitinib-resistance in certain cancers, such as non-small cell lung cancer (NSCLC) cells [37]. Confocal microscopy revealed that both NDAT and PI3K inhibitor, LY294002 inhibited the abundance of PD-L1 protein in the established HT-29 CRC cell and three established primary cultures of CRC cells (Colo_160224, Colo_150624, and Colo_160426) (Figure 3A). The gene expression and protein accumulation of PD-L1 suppressed significantly by NDAT and the PI3K inhibitor, LY294002, were demonstrated in Colo_150624 primary CRC cell culture (Figure 3B and 3C). These results suggested that blockage of PI3K activity could inhibit PD-L1 accumulation. NDAT also demonstrated its anti-PI3K activity [9]. Furthermore, the PI3K inhibitor, LY294002, induced anti-proliferation in gefitinib-resistant primary CRC cells, Colo_160224 primary CRC cell cultures (Figure 4A).

To investigate the effect of PI3K activation in the NDAT- or gefitinib-induced anti-proliferation in gefitinib-resistant primary CRC cells, Colo_160224 primary CRC cell cultures were treated with 0.1 μM NDAT, 10 µM gefitinib, 10 µM LY294002, or their combination in the medium with reagents changed daily for six days. Both NDAT and LY294002 inhibited cell proliferation significantly (Figure 4B). On the other hand, gefitinib stimulated cell proliferation significantly (Figure 4B), as observed in Figure 1. The combined NDAT and gefitinib or gefitinib in the presence of LY294002 inhibited cell growth significantly (Figure 4B).

To address the effect of combinations of agents on gefitinib-resistant CRC proliferation, we examined the effects of NDAT, LY294002, or gefitinib and their combination on the expression of *PD-L1* and related genes in Colo_160224 cells (Figure 4C). After 24 h of treatment, cells were harvested, and RNA was extracted. The qPCR assay was conducted for *PD-L1*, *CCND1* (proliferation), *CASP2* (anti-proliferation), and *MMP-9* (metastasis). NDAT significantly suppressed the expression of *PD-L1*, *CCND1*, and *MMP-9*, and enhanced that of *CASP2* in Colo_160224 cells, as we have reported previously in *K-RAS* mutant CRC HCT116 cells [9,10,12]. LY294002 significantly reduced the expression of *PD-L1* and *CCND1*, but not that of *CASP2* and *MMP-9*. However, gefitinib stimulated the expression of *PD-L1* and *MMP-9* and suppressed *CASP2* expression. The combined treatment of NDAT with gefitinib showed a significantly potentiated effect on *CASP2* expression and inhibited more significantly the expression of *PD-L1*, *CCND1*, and *MMP-9*. The combination of LY294002 with gefitinib reduced the expression of *CCND1* significantly as compared untreated control or gefitinib treatment alone, although the effect was less than LY294002 treatment alone (Figure 4C). The combined treatment of LY294002 with gefitinib inhibited *PD-L1* expression significantly as compared with gefitinib treatment alone. Also, the same phenomenon appeared in the combined treatment of NDAT and gefitinib.

### 3.4. NDAT Inhibits PD-L1 Accumulation and Tumor Growth in HCT116 Cell Mouse Xenografts 

In order to confirm our observation that NDAT suppressed PI3K to downregulate PD-L1 accumulation and cell proliferation in gefitinib-resistant CRC cells, xenograft of gefitinib-resistant HCT116 cells was conducted. The tumor size was assessed in BALB/C nude mice with gefitinib and NDAT treatment (Appendix A). NDAT had a more significant inhibitory effect on tumor growth than gefitinib. Then immunohistochemical staining of PD-L1 was apparently primarily on the outer cells of the tumor masses at five weeks. Expression of PD-L1 was decreased in xenografts of mice treated with gefitinib (10 mg/kg) and in the tumors of mice that received the combination of NDAT (1 mg/kg) and gefitinib (10 mg/kg) (Figure 5).

In a previous study, we demonstrated that NDAT enhanced the anti-proliferative activity of gefitinib in gefitinib-resistant CRC cells by inhibiting ST6Gal1 activity and PI3K activation. NDAT also enhanced the anticancer activity of gefitinib in HCT116 CRC cell xenografts [9]. In the current experiments, we further investigated the histological changes in the HCT116 cell-derived xenograft tissues after these treatments of NDAT, gefitinib, and their combination. The morphological changes of cancer cells with various treatments in xenografts were shown in Figure 6. Typical tumor morphology, such as a high nucleus/cytoplasm ratio and active mitosis, was observed in tissues from the control xenografts. The structure of the tumor masses was loosened, and the tumor cells were not packed together tightly after NDAT treatment. However, this was not observed in the gefitinib-treated group. The combination of NDAT (1 mg/kg) with gefitinib (10 mg/kg) showed destruction of tumor masses, disordered cell morphology, and extensive necrosis characterized by pyknosis. As a result, the non-tumor space was occupied by connective tissue. Reagent-treated groups reduced the nucleus-to-cytoplasm ratio compared to the control group. The combination of NDAT (1 mg/kg) and gefitinib (10 mg/kg) had less compact tumor mass and more connective tissue filling.

qPCR studies were carried out in tumor samples harvested from xenografts to confirm the suppressive effects of NDAT on PI3K-related signal transduction pathway in gefitinib-resistant CRC. The *PI3K* expression was inhibited by NDAT. Alternatively, gefitinib stimulated *PI3K* expression. However, the combination of NDAT and gefitinib inhibited *PI3K* expression significantly (Figure 7). Additionally, the expression of *PD-L1*, reduced by gefitinib, was further downregulated by NDAT co-administration. Also, NDAT significantly increased the expression of the pro-apoptotic gene *p53* and reduced the expression of *CCND1*, *VEGF-A*, and *bFGF* but these were not observed in gefitinib-treated animals (Figure 7). Western blot analyses showed that NDAT but not gefitinib inhibited PD-L1 accumulation significantly in HCT116 xenograft tissues (Figure 8). In summary, although gefitinib may affect *PD-L1* expression, NDAT inhibited *PI3K* expression and PD-L1 accumulation to inhibit cell proliferation in gefitinib-resistant CRC proliferation in vitro and in vivo.

## 4. Discussion

PD-1 belongs to the cluster of differentiation (CD) 28/CTL-associated antigen (CTLA)-4/inducible T cell costimulator (ICOS) family. It is categorized as an inhibitory receptor expressed by T cells, dendritic cells, natural killer cells, macrophages, and B cells [38]. PD-1-mediated T cell signaling depends upon binding of its ligands, PD-L1 (B7-H1) or PD-L2 (B7-DC) to alter cytotoxic killing, cytokine production, and T cell proliferation [39]. Expression of PD-L1 and PD-L2 are up-regulated in many human solid tumors, including CRC [14,40,41]. Blockade of PD-1 or PD-L1 activities may provide an optional management approach for PD-L1-overexpressing tumors, e.g., NSCLC with EGFR mutation, that has EGFR-TKI resistance [19]. Recently, we demonstrated that NDAT non-immunologically downregulates both gene expression and protein accumulation of basal and T_4_-induced PD-L1 in cancer cells, such as oral cancer cells, breast cancer cells, and CRC cells [10,12].

In this study, NDAT inhibited protein accumulation of PD-L1 in a concentration-dependent manner. In addition, gefitinib decreased the protein accumulation of PD-L1 in HCT116 cell xenografts. This finding is consistent with previous reports that gefitinib decreased PD-L1 expression in NSCLC cell lines [42,43]. The combination of NDAT and gefitinib further inhibited PD-L1 protein accumulation in xenografts. The PD-1/PD-L1 axis can modulate nature killer cells’ actions on tumor cells [44,45]. Gefitinib was reported to immuno-regulate lung cancer cells and NK cells that could enhance the interaction between them [43]. As a result, the combination of NDAT and gefitinib in our studies showed enhanced action on tumor size reduction in HCT116 cell xenografts [9].

The combination of NDAT and gefitinib also reduced viability and inhibited expression of *PD-L1* in primary cultures of human CRC cells. One of the mechanisms for tumor cell-stimulated expression of PD-L1 is the activation of the EGFR, ERK1/2, PI3K-Akt, or Janus kinase 2/STAT1 signaling pathways [46]. In our studies, gefitinib did not inhibit the activation of PI3K but instead activated it in one primary culture (Colo_160224 cells). NDAT not only reduced PD-L1 expression but also inhibited PI3K activate by gefitinib (Appendix A). As a result, gefitinib (10 μM) induced the gene expression of *PD-L1* in these cells (Figure 2D) and improved the viability in these cells (Figure 1D). On the other hand, NDAT not only inhibited the activation of PI3K but also decreased the expression of *PI3K*. Thus, PI3K appears to play an essential role in the inhibitory effect on NDAT-blocked PD-L1 expression in CRC cells.

Patients with phosphorylated EGFR-dependent cancer cell proliferation and metastasis are well-suited for therapy with EGFR TKIs [47]. However, the mutation of EGFR or downstream signaling pathways linked to EGFRs reduce the efficacy of such targeting therapy. Gefitinib is the first generation of EGFR-TKI to target EGFR-19del and EGFR-L858R mutation [19], but it has been less effective in CRC management than in other types of cancer [48,49,50]. In contrast to its efficacy in NSCLC treatment, gefitinib used in CRC patients in Phase I trials attained stable disease without an objective response in tumor size. This was the case although it may be effective in higher doses for lung cancer patients [48,51]. However, gefitinib, in combination with other EGFR-targeted agents, has been found to have antitumor effects that were more impressive than those obtained with the administration of a single EGFR inhibitors [52].

Atorvastatin has been shown to enhance its anticancer effects in combination with gefitinib. At a dose of 5 μM, atorvastatin enhanced the anticancer activity of gefitinib, apparently through concomitant inhibition of AKT and ERK signaling activities [53]. In current preclinical studies, NDAT improved the anti-proliferative effectiveness of gefitinib in primary cultures of CRC cells. The combined use of these two agents in studies of primary human CRC cells suppressed the transcription of genes regarded as biomarkers of proliferation, anti-proliferation, and angiogenesis.

In gefitinib-resistant HCT116 xenograft studies, NDAT suppressed the expression of *PI3K* and *PD-L1* (Figure 7). On the other hand, gefitinib only suppressed the expression of PD-L1 but not PI3K (Figure 7). However, their combination further downregulated the expression of *PD-L1* and *PI3K* (Figure 7). These observations suggested that NDAT could compensate gefitinib-regulated gene expression in gefitinib-resistant CRC cells. The accumulation of PD-L1 in xenograft studies further confirmed NDAT but not gefitinib-reduced PD-L1 accumulation (Figure 8). Although it is not clear what mechanisms are involved in gefitinib-resistance in Colo_160224 primary cell culture; however, NDAT suppressed PI3K activation and PD-L1 accumulation to inhibit cancer growth in Colo_160224 cells and *K-RAS* mutant HCT116 CRC cells.

## 5. Conclusions

NDAT was shown to enhance the anti-proliferative effects of gefitinib in four primary cultures of human CRC cells. In all four cell lines, NDAT alone suppressed *PD-LI* expression and proliferation. The combination treatment of NDAT and gefitinib further enhanced this effect. However, different mechanisms were involved in gefitinib- and NDAT-induced inhibition of PD-L1 accumulation. Gefitinib-inhibited PD-L1 accumulation was not inactive PI3K-dependent; however, NDAT inhibited PD-L1 accumulation via PI3K inactivation. These results affirm certain anticancer actions of NDAT and suggest that the combination of NDAT and gefitinib may be useful clinically in the setting of human gefitinib-resistant CRC.

## Figures and Tables

**Figure 1 cells-09-01830-f001:**
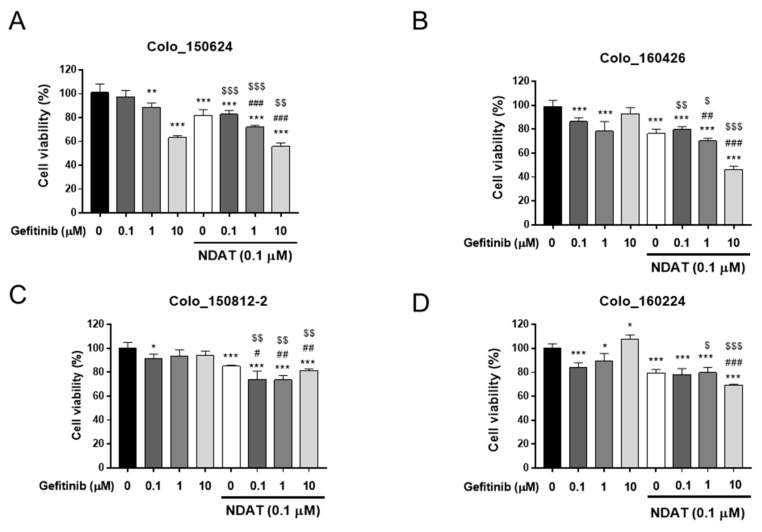
NDAT augments gefitinib-stimulated anti-proliferation in primary cultures of human CRC cells. Four established primary human CRC cell cultures, Colo_150624 (**A**), Colo_160426 (**B**), Colo_150812-2 (**C**), and Colo_160224 (**D**), were seeded in 96-well plates and treated with different concentrations of gefitinib (0.1, 1, and 10 μM), NDAT (0.1 μM), or their combination. Media with drugs were refreshed daily for six days. Cell viability was examined with the MTT assay. *N* = 6. Data are expressed as mean ± SD; * *p* < 0.05, ** *p* < 0.01, *** *p* < 0.001, compared with untreated control; ^#^
*p* < 0.05, ^##^
*p* < 0.01, ^###^
*p* < 0.001, compared with NDAT; ^$^
*p* < 0.05, ^$$^
*p* < 0.01, ^$$$^
*p* < 0.001, compared with gefitinib.

**Figure 2 cells-09-01830-f002:**
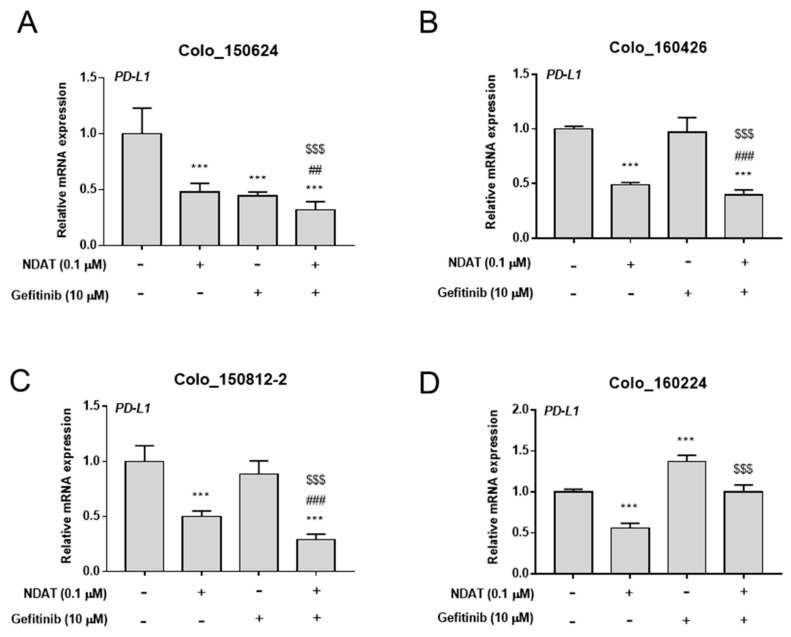
NDAT attenuate expression of *PD-L1* regulated by gefitinib in human CRC primary cell cultures. Four established primary human CRC cell cultures, Colo_150624 (**A**), Colo_160426 (**B**), Colo_150812-2 (**C**), and Colo_160224 (**D**), were seeded in six-well plates and treated with different concentrations of gefitinib (10 μM), NDAT (0.1 μM), or their combination after starvation for 24 h. Cells were harvested and total RNA was extracted. qPCR was conducted for *PD-L1* expression. *N* = 6. Data are expressed as mean ± SD; *** *p* < 0.001, compared with untreated control; ^###^
*p* < 0.001, compared with NDAT; ^$$$^
*p* < 0.001, compared with gefitinib.

**Figure 3 cells-09-01830-f003:**
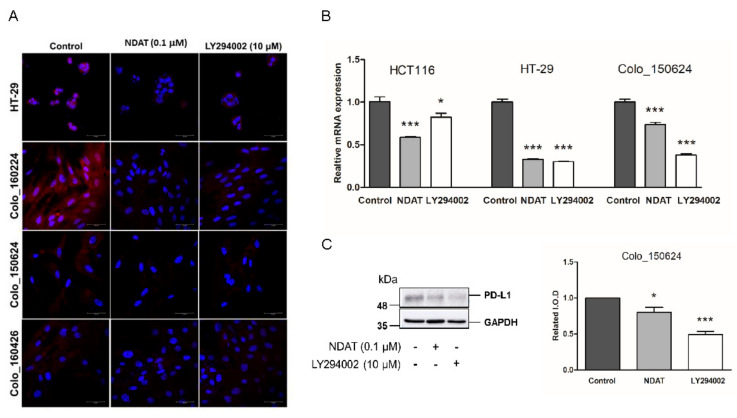
NDAT and LY294002 inhibit PD-L1 accumulation in CRC. HT-29 cells (colorectal cancer cell line) and three established primary human CRC cell cultures (Colo_160224, Colo_150624, and Colo_160426) were treated with NDAT or LY294002 for 24 h. (**A**) Immunocytochemistry was performed for PD-L1 (red) and the nucleus was counterstained with DAPI (blue). Scale bar = 50 µm. (**B**) Gene expression of *PD-L1* was evaluated by qPCR. (**C**) Protein accumulation of PD-L1 was detected by western blot. Densitometric analyses of western blots were done by ImageJ 1.5 and shown in a bar chart. *N* = 3. Data are expressed as mean ± SD; * *p* < 0.05, *** *p* < 0.001, compared with the untreated control.

**Figure 4 cells-09-01830-f004:**
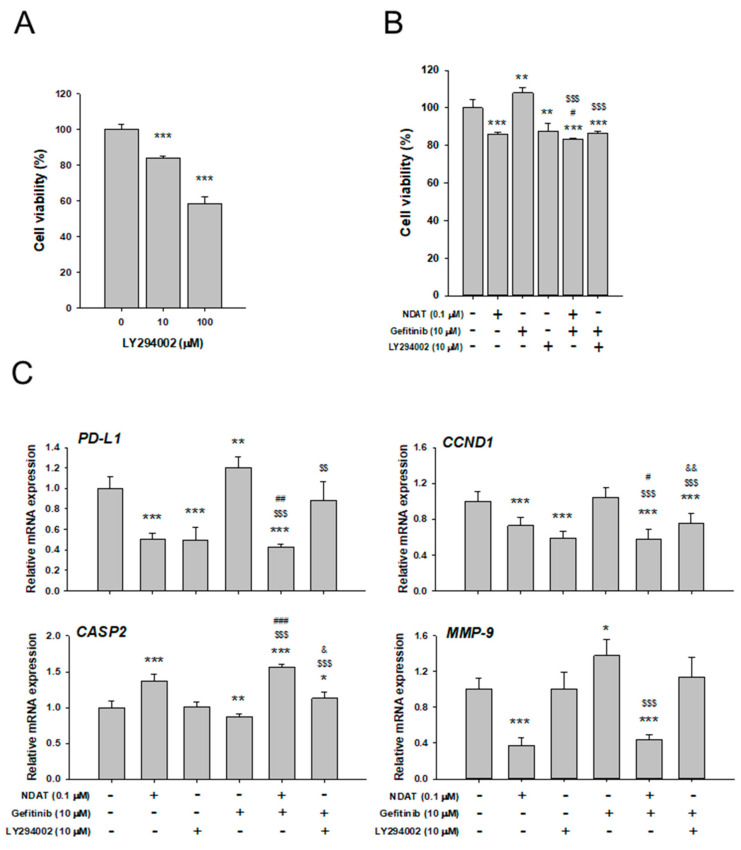
Actions of NDAT and gefitinib are PI3K-dependent in primary cell culture Colo_160224. (**A**) Cells were treated with LY294002 (10 and 100 μM) with refreshed media containing LY294002 daily for six days. Cell viability was conducted by the MTT assay. *N* = 6. (**B**) Cells were treated with NDAT (0.1 μM), gefitinib (10 μM), and their combination with or without LY294002 (10 μM) for six days. Cell viability was evaluated by the MTT assay. *N* =6. (**C**) Cells were treated with NDAT (0.1 μM), gefitinib (10 μM), and their combination with or without LY294002 (10 μM) after starvation for 24 h (*N* = 4). Total RNA was extracted, and qPCR was performed for *PD-L1*, *CCND1*, *CASP2*, and *MMP-9*. Data are expressed as mean ± SD; * *p* < 0.05, ** *p* < 0.01, *** *p* < 0.001, compared with the untreated control; ^#^
*p* < 0.05, ^##^
*p* < 0.01, ^###^
*p* < 0.001, compared with the NDAT; ^$$^
*p* < 0.01, ^$$$^
*p* < 0.001, compared with gefitinib; ^&^
*p* < 0.05, ^&&^
*p* < 0.01, compared with LY294002.

**Figure 5 cells-09-01830-f005:**
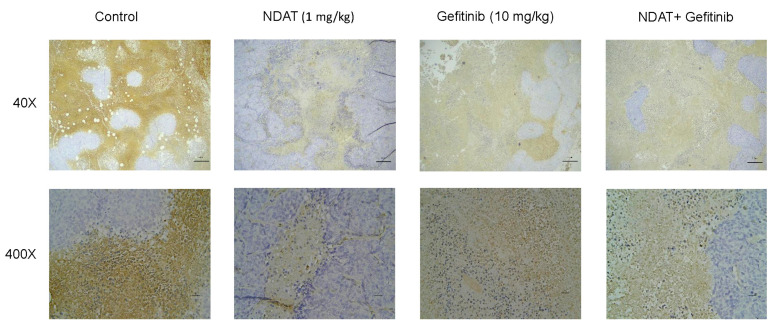
NDAT potentiates gefitinib-reduced PD-L1 expression in HCT116 cell-derived xenograft tumors (Table 1). The positive staining of PD-L1 is shown in brown color. The magnification is 40× (upper panels, bar = 210 μm) and 400× (lower panels, bar = 21 μm).

**Figure 6 cells-09-01830-f006:**
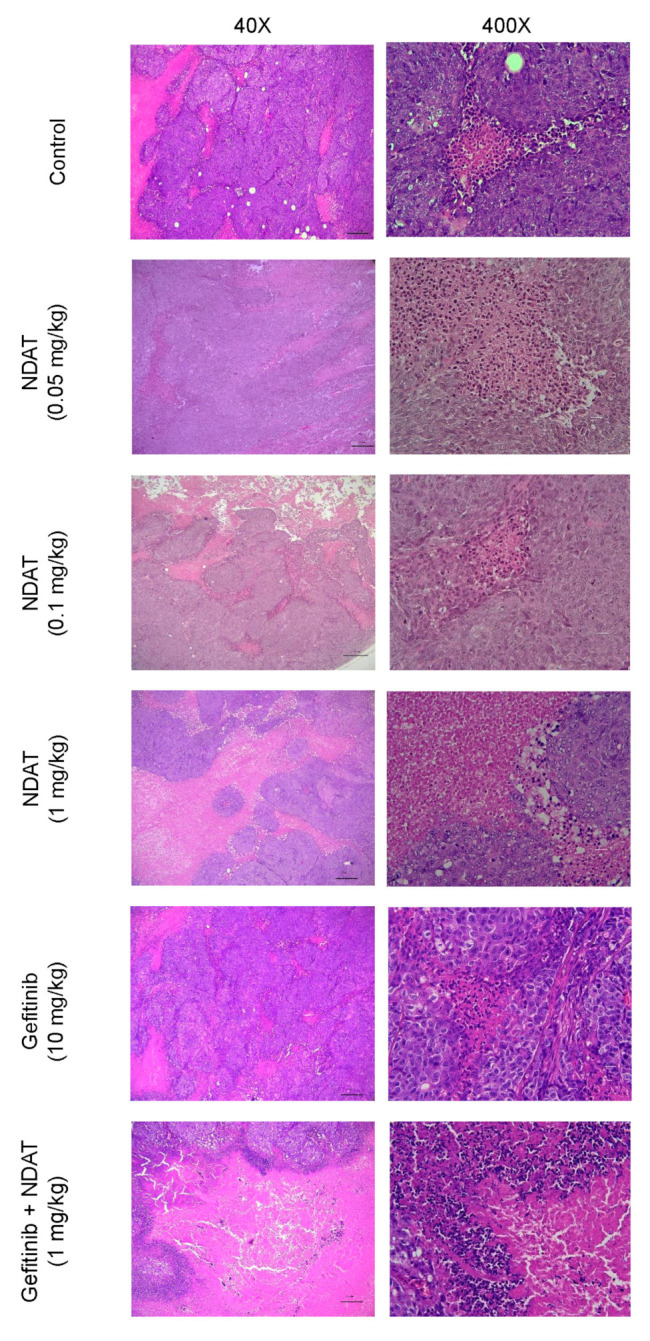
H&E stain of HCT116 xenograft tumor tissues. Xenograft tissues from various treatment including NDAT (0.05 mg, 0.1 mg, 1 mg/kg), gefitinib (10 mg/kg) or combined treatment were stained by hematoxylin (nucleus, blue) and eosin (extracellular matrix and cytoplasm, pink). The magnification is 40× (left panels, bar = 210 μm) and 400× (right panels, bar = 21 μm).

**Figure 7 cells-09-01830-f007:**
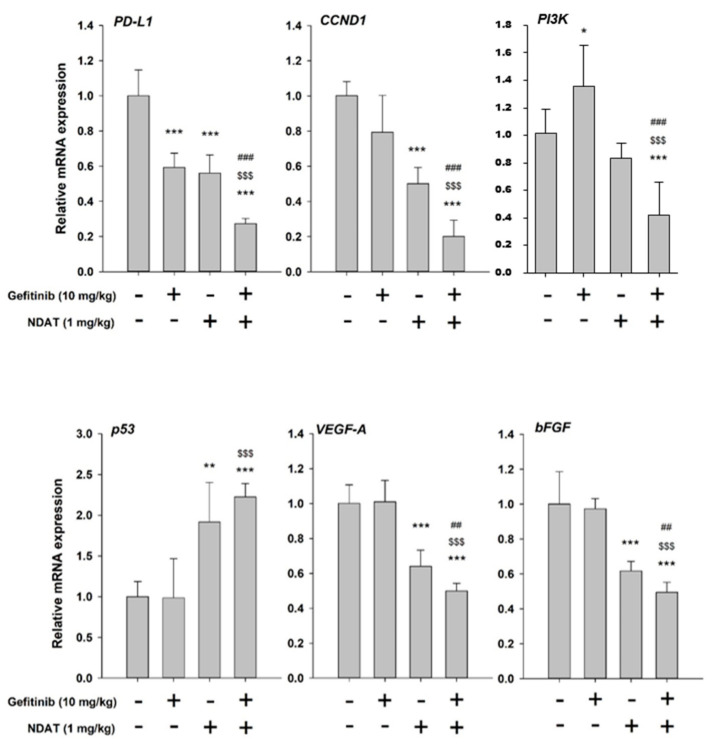
NDAT enhances gefitinib-regulated gene expression in HCT116 cell-derived xenograft tumor tissues. Tumor tissues were homogenized, and total RNA was extracted. Total RNA was then transcribed into cDNA. The expression levels of *CCND1, p53, VEGF-A, PD-L1, EGFR*, and *bEGF (EGF2)* were examined with qPCR experiments. The number of independent experiments *N* = 4. Data are expressed as mean ± SD; * *p* < 0.05, ** *p* < 0.01, **** p* < 0.001, compared with untreated controls; ^##^
*p* < 0.01, ^###^
*p* < 0.001, compared with NDAT; ^$$$^
*p* < 0.001, compared with gefitinib.

**Figure 8 cells-09-01830-f008:**
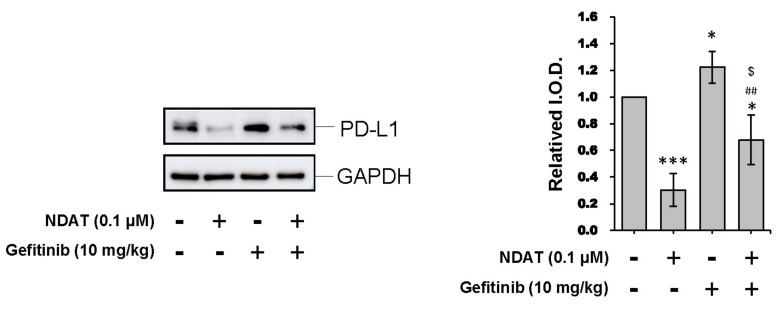
NDAT inhibited PD-L1 accumulation in HCT116 cell-derived xenograft tumor tissues. Tumor tissues were homogenized, and total proteins were extracted. Western blotting analyses were conducted by PD-L1. The number of independent experiments *N* = 4. Data are expressed as mean ± SD; * *p* < 0.05, *** *p* < 0.001, compared with untreated control; ^##^
*p* < 0.01, compared with NDAT; ^$^
*p* < 0.05, compared with gefitinib.

**Table 1 cells-09-01830-t001:** Primer Sequences for Quantitative PCR on CRC Cell Lines and Primary Culture Cells.

NAME	F―ATCG	R―ATCG
**18S**	5’-GTAACCCGTTGAACCCCATT-3’	5’-CCATCCAATCGGTAGTAGCG-3’
**PD-L1**	5’-GTTGAAGGACCAGCTCTCCC-3’	5’-ACCCCTGCATCCTGCAATTT-3’
**CCND1**	5’-CAAGGCCTGAACCTGAGGAG-3’	5’-GATCACTCTGGAGAGGAAGCG-3’
**CASP2**	5’-GCATGTACTCCCACCGTTGA-3’	5’-GACAGGCGGAGCTTCTTGTA-3’
**MMP-9**	5’-TGTACCGCTATGGTTACACTCG-3’	5’-GGCAGGGACAGTTGCTTCT-3’
**PI3K**	5’-CCTGATCTTCCTCGTGCTGCTC-3’	5’-ATGCCAATGGACAGTGTTCCTCTT-3’
**bFGF**	5’-GAGAAGAGCGACCCTCACA-3’	5’-TAGCTTTCTGCCCAGGTCC-3’
**VEGF-A**	5’-TACCTCCACCATGCCAAGTG-3’	5’-GATGATTCTGCCCTCCTCCTT-3’
**p53**	5’-AAGTCTAGAGCCACCGTCCA-3’	5’-CAGTCTGGCTGCCAATCCA-3’

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
