# Peer review of "NDAT Targets PI3K-Mediated PD-L1 Upregulation to Reduce Proliferation in Gefitinib-Resistant Colorectal Cancer"

_cells, 2020, doi:10.3390/cells9081830_

Round 1

Reviewer 1 Report

It is a well-prepared manuscript describing NDAT could enhance anti-proliferative effect of gefitinib in CRC, and exploring its mechanism related to gefitinib- and NDAT-induced inhibition of PD-L1 accumulation.  I have only a minor comment. Why the author choose HCT 116 as a model for in vitro and in murine xenografts study, what about other KRAS mutations in CRC? Fig 5, 6, it is suggested to show higher magnification images.

Author Response

Reviewer 1

I have only a minor comment. Why the author choose HCT 116 as a model for in vitro and in murine xenografts study, what about other KRAS mutations in CRC? Fig 5, 6, it is suggested to show higher magnification images.

Thanks for the review’s comment. We have published a related paper entitled” Enhancement by Nano-Diamino-Tetrac of Antiproliferative Action of Gefitinib on Colorectal Cancer Cells: Mediation by EGFR Sialylation and PI3K Activation” in Horm Cancer. 2018 Dec; 9 (6):420-432. In that paper, studies were conducted in HCT116 and HT-29 cells. Because HCT116 cells are gefitinib-resistant and the current paper continued previous work, therefore, we used HCT116 cells. In response to the reviewer’s comments, we re-took photographs in Figure 5 and 6.

Reviewer 2 Report

The authors have tested efficacy of L-Thyroxine analouge -nanoparticulate analogue nano-diamino-tetrac (NDAT) in gefitinib resistant colorectal cell lines. In the present study they explored the mechanism by which NDAT enhances antiproliferative effect of gefitinib in CRC. They performed MTT assay to identify changes in cell proliferation upon treatment with or without NDAT at different conc. qPCR, western blotting, immunostaining was used to identify changes in PDL-1 expression with different treatments in-citro an in-vivo. They observed that combination treatment of NDAT and gefitinib  inhibited PD-L1 expression and cell proliferation and that NDAT reduced PI3K expression, PD- L1 accumulation and in-vivo tumor growth in HCT116 (K-RAS mutant).  The authors conclude that gefitinib may suppress PD-L1 expression but did not inhibit proliferation through PI3K in gefitinib resistant primary CRC cells, while NDAT not only down-regulated PD-L1 expression via 65 blocking PI3K activation but also inhibited cell proliferation in gefitinib-resistant CRCs.

Few minor concerns in the article are as follows:

  1. Fig. 1 Please indicate if the authors included a vehicle control for NDAT to counter any non-specific effect from nanoparticles?
  2. Fig. 2 Please increase the font of text in x-axis of the graph.
  3.  Fig. 3 The title indicates changes accumulation of PDL-1 with NDAT and PI3K inhbitor, while the authors show change in protein expression. Pleae clarify the title or the results carefully and include proper text. The basal expression of PDL-1 looks very low in the tested cell lines (No membrane staining is observed for PDL-1 in the figure). Please include a western blot to show changes in PDL-1 expression upon treatment with NDAT or PI3K inhibitor.
  4. Fig.5 Please include a higher magnification image to clearly show surface staining for PDL-1
  5. Fig. 8 Please label the lanes of the western blot to indicate tested samples.

Author Response

Reviewer 2

Few minor concerns in the article are as follows:

  1. 1 Please indicate if the authors included a vehicle control for NDAT to counter any non-specific effect from nanoparticles?

Thanks for the review’s comment. NDAT is dissolved in PBS, therefore, it doesn’t affect the cell viability.

  1. 2 Please increase the font of text in x-axis of the graph.

Thanks to the reviewer’s comment. The font size was increased in Figure 2 of the revised manuscript.

  1. 3 The title indicates changes accumulation of PD-L1 with NDAT and PI3K inhibitor, while the authors show change in protein expression. Please clarify the title or the results carefully and include proper text. The basal expression of PD-L1 looks very low in the tested cell lines (No membrane staining is observed for PD-L1 in the figure). Please include a western blot to show changes in PD-L1 expression upon treatment with NDAT or PI3K inhibitor.

Thanks to the reviewer’s comment. Western blot of the inhibitory effect of NDAT or PI3K inhibitor was presented. NDAT inhibited inducible PD-L1 expression and protein accumulation by the inhibition of activated ERK1/2 and PI3K (Food Chem Toxicol. 2018 Oct;120:1-11. doi: 10.1016/j.fct.2018.06.058).

  1. 5 Please include a higher magnification image to clearly show surface staining for PD-L1.

Thanks to the reviewer’s comment. We have included the magnified images in Figure 5. PD-L1 in our studies presents in cytosol as well as on cell surface. (Endocr Relat Cancer. 2018 May;25(5):533-545. doi: 10.1530/ERC-17-0376).

  1. 8 Please label the lanes of the western blot to indicate tested samples.

Thanks to the reviewer’s comment. It is our mistake not adding lane labels on western blot. The lane labels of Figure 8 were added to the revised manuscript.

Reviewer 3 Report

The manuscript by Huang et al., describes the effect of NDAT in enhancing anti-proliferative effect in Gefitinib resistant colorectal cancer. Although the manuscript is a comprehensive effort with good experiments, it needs major revision to significantly increase readability and clarity in reporting the results, figures and interpretation of the results. The results are not convincing enough to make sound conclusion about the combination effect of Gefitinib and NDAT.

Major comments:

  1. Fig 1 needs major overhaul for easy understanding. I don’t see any point in showing 2 different combinations in the main figure. NDAT (10^-8 M/ 10 nM) combinations can be moved to supplementary figs and please change the axis to uM or nM.
  2. Fig 1: Separate the single agent Gefitinib bars from the combination bars either with an extra space or a dotted- vertical line. The notations for P- value also make the graph busy and confusing. Prioritize which significant effect is being explained in the text and highlight only those P value on the bar graph.
  3. Fig 1: The single agent effect of NDAT is not shown in this graph which is important to interpret the results better. From the graph, I would interpret the results as: Other than Colo_150624, the other 3 lines develop resistance to Gefitinib at different concentrations and the combination of Gefitinib and NDAT sensitizes 2/3 resistant lines (Colo_160426 and Colo_160224).
  4. Figure 2: Please move fig 2A to the supplementary figs. Change the Y-axis to fold change instead of relative abundance to make the graph meaningful. The report and discussion of the results is unnecessarily complicated. From the graph, it is clear that the three resistant lines show decreased PD-L1 expression after combination with NDAT, with a significant increase in PDL1 upregulation with gefitinib alone in Colo_160224 line. Since 1uM gefitinib alone and in combination shows similar effect in all 4 lines, it can be moved to supplementary figs as well.
  5. Figure 3: In addition to immunofluorescence, qPCR or western should be done to confirm the finding and also to validate target inhibition with LY294002. Since 160426 and 150624 samples barely have any PD-L1 expression, it is hard to convince the readers on the inhibitory effect.
  6. Figure 4: The effect of NDAT and Gefitinib combination in cell viability is not convincing. Please report the 'fold decrease' compared to the untreated controls and single agent.
  7. For in vivo xenograft studies, The tumor growth curve is not reported and confirmation of target inhibition with single agents is also not reported. Without the anti-proliferative effect or reduction of tumor growth in vivo, the effect of combination is not convincing.

Minor Comments:

  1. The title can be re-worded to be more specific: “NDAT targets PI3K mediated PD-L1 upregulation to reduce proliferation in Gefitinib resistant colorectal cancer”
  2. The way of reporting concentrations of NDAT should be comparable with Gefitinib for better readability. Please report 10^-7 M as 100 nM and so on, in all the figures.
  3. In methods section for cell viability assay, please report the assay duration in days and not hours for long term treatment.
  4. Make a supplementary table for the primer sequences and Accession numbers
  5. If possible, please make the primary colorectal samples nomenclature easy, if being reported for the first time.
  6. Splitting the Y-axis in figures seems unnecessary in most cases. Use it judiciously (Fig 1C, 1D, 4B)
  7. Figure 6 with H&E stain can be moved to supplementary since it corroborates previous finding.
  8. Figure 8 WB does not have any labels.

Author Response

Reviewer 3

Major comments:

  1. Fig 1 needs major overhaul for easy understanding. I don’t see any point in showing 2 different combinations in the main figure. NDAT (10^-8 M/ 10 nM) combinations can be moved to supplementary figs and please change the axis to uM or

Thanks to the reviewer’s comments. It has been a long time to present hormone and analog used molar. It was used μM through all figures in the revised manuscript and those combined labeled figures (NDAT 0.01 μM) were presented in the supplementary Figure S1.

  1. Fig 1: Separate the single agent Gefitinib bars from the combination bars either with an extra space or a dotted- vertical line. The notations for P- value also make the graph busy and confusing. Prioritize which significant effect is being explained in the text and highlight only those P value on the bar graph.

Thanks to the reviewer’s comments. Figure 1 was reblotted as suggested in the revised manuscript.

  1. Fig 1: The single agent effect of NDAT is not shown in this graph which is important to interpret the results better. From the graph, I would interpret the results as: Other than Colo_150624, the other 3 lines develop resistance to Gefitinib at different concentrations and the combination of Gefitinib and NDAT sensitizes 2/3 resistant lines (Colo_160426 and Colo_160224).

Thanks to the reviewer’s comments. Treatment of NDAT was shown in Figure 1 as with 0 µM gefitinib. Except for Colo_150812-2, all other three primary CRC cell lines were sensitized to NDAT treatment at 0.01 μM (10-8 M) as shown in the primary submitted manuscript. Now we moved the result of combined treatment of lower NDAT dosage (0.01 μM) to Figure S1.

  1. Figure 2: Please move fig 2A to the supplementary figs. Change the Y-axis to fold change instead of relative abundance to make the graph meaningful. The report and discussion of the results is unnecessarily complicated. From the graph, it is clear that the three resistant lines show decreased PD-L1 expression after combination with NDAT, with a significant increase in PDL1 upregulation with gefitinib alone in Colo_160224 line. Since 1µM gefitinib alone and in combination shows similar effect in all 4 lines, it can be moved to supplementary figs as well.

Thanks to the reviewer’s comments. For Figure 2A, we have replaced relative abundance with fold change and then moved the bar chart to Figure S2A. The gefitinib treatment (1 μM) in Figure 2B to 2E was moved to the supplementary Figure S2 in the revised manuscript. There were significant differences compared between gefitinib and the combined treatment thus these figures were kept intake.

  1. Figure 3: In addition to immunofluorescence, qPCR or western should be done to confirm the finding and also to validate target inhibition with LY294002. Since 160426 and 150624 samples barely have any PD-L1 expression, it is hard to convince the readers on the inhibitory effect.

Thanks to the reviewer’s comments. We totally agree with the reviewer's suggestion. The qPCR was conducted to confirm the inhibitory effect of LY294002 on PD-L1 expression in two established CRC cell lines and one primary CRC cell line (Colo_150624) in Figure 3B. Furthermore, we assessed the protein expression of PD-L1 in Colo_150624 (Figure 3C). Since Colo_160426 cell has been used up in the previous experiment, only the results of Colo_150624 can be presented.

  1. Figure 4: The effect of NDAT and Gefitinib combination in cell viability is not convincing. Please report the 'fold decrease' compared to the untreated controls and single agent.

Thanks to the reviewer’s comments. For the consistency of the analysis in cell viability experiment, Figure 4A in the revised version is still expressed as a percentage.

  1. For in vivo xenograft studies, the tumor growth curve is not reported and confirmation of target inhibition with single agents is also not reported. Without the anti-proliferative effect or reduction of tumor growth in vivo, the effect of combination is not convincing.

Thanks to the reviewer’s comments. The inhibitory effect of tumor growth by gefitinib, NDAT, and the combination was presented in Figure S3 in the revised manuscript. We have added a description at line 321 in the revised manuscript.

Minor Comments:

  1. The title can be re-worded to be more specific: “NDAT targets PI3K mediated PD-L1 upregulation to reduce proliferation in Gefitinib resistant colorectal cancer

Thanks to the reviewer’s comments. The title of the reviewer’s suggestion is indeed more specific for our findings. Therefore, we changed the title form “NDAT Targets PI-3K-PD-L1 to Induce Anti-Proliferation in Gefitinib-Resistant Colorectal Cancer” to “NDAT targets PI3K-mediated PD-L1 upregulation to reduce proliferation in Gefitinib resistant colorectal cancer”

  1. The way of reporting concentrations of NDAT should be comparable with Gefitinib for better readability. Please report 10^-7 M as 100 nM and so on, in all the figures.

Thanks to the reviewer’s comments. We have changed 10-7 M to 0.1 μM in all the figures.

  1. In methods section for cell viability assay, please report the assay duration in days and not hours for long term treatment.

Thanks to the reviewer’s comments. We have replaced hours with days. The corrected words are list in revised manuscript: line 144, 233, 286, 309 and 311.

  1. Make a supplementary table for the primer sequences and Accession numbers

Thanks to the reviewer’s comments. We agreed with the reviewer's suggestion and included a table of qPCR primers (Table 1) in the methods section of the revised manuscript.

  1. If possible, please make the primary colorectal samples nomenclature easy, if being reported for the first time.

Thanks to the reviewer’s comments. For the compilation of laboratory data, we cannot simplify the nomenclature of primary colorectal cells. Thus, the name of primary colon cells is kept in revised manuscript.

  1. Splitting the Y-axis in figures seems unnecessary in most cases. Use it judiciously (Fig 1C, 1D, 4B)

Thanks to the reviewer’s comments. We have corrected the display of the Y-axis scale and the split bar graph.

  1. Figure 6 with H&E stain can be moved to supplementary since it corroborate previous finding.

Thanks to the reviewer’s comments. Although H&E stain confirms the previous phenomenon, there may be differences in cell and tissue between H&E stain and in vitro experiments. This is an important result for our finding, thus we decided to put it in the manuscript.

  1. Figure 8 WB does not have any labels.

Thanks to the reviewer’s comments. It is our mistake not adding lane labels on western blot. We have added the experimental condition in Figure 8.

Round 2

Reviewer 3 Report

The authors have sufficiently addressed all the concerns raised by all reviewers and I recommend the manuscript to be accepted in the present form to be published.